# Effects of Fermented Herbal Tea Residue on Serum Indices and Fecal Microorganisms of Chuanzhong Black Goats

**DOI:** 10.3390/microorganisms10061228

**Published:** 2022-06-16

**Authors:** Chongya Gao, Longfei Wu, Weiran Zhao, Yiye Chen, Ming Deng, Guangbin Liu, Yongqing Guo, Baoli Sun

**Affiliations:** College of Animal Science, South China Agricultural University, Guangzhou 510642, China; a15521139942@163.com (C.G.); longfeiwu2020@163.com (L.W.); a745997401@163.com (W.Z.); chenyiye1234@126.com (Y.C.); dengming@scau.edu.cn (M.D.); gbliu@scau.edu.cn (G.L.)

**Keywords:** fermented herbal residue, goats, microorganism, serum biochemical indices, fermented feed

## Abstract

Herbal tea residues (HTRs) are a by−product of herbal tea processing that contains many nutrients and active substances but are often discarded as waste. The main aim of the present study was to determine the food safety of HTRs and lay the foundation for its use as a novel feed resource for goats. In this study, discarded HTRs were fermented and then fed to 33 female Chuanzhong black goats (121 ± 4.00 days) with similar weight (9.33 ± 0.95 kg) and genetic background, which were divided into three groups (fermented herbal tea residue (FHTR) replacement of 0%, 15% and 30% of the forage component of the diet). The feeding experiment lasted for 35 days. On day 35, our findings indicated that the concentrations of hydroxyl radicals and urea increased linearly, and the concentrations of glutathione peroxidase increased quadratically with the increase in FHTR. In addition, we investigated the fecal microbiota composition of eight Chuanzhong black goats in the control, 15% and 30% FHTR replacement groups and found that FHTR had no remarkable effect on the fecal microbiota composition. Results indicated that goat physiological functions remained stable after FHTR was added to the diet.

## 1. Introduction

Herbal tea (HT), which originated in Guangdong, Fujian, and other regions in southeast China, is an oral liquid made from a variety of herbs. It has different functions, such as clearing away heat, detoxifying, producing body fluid, quenching thirst, dispelling fire and dehumidifying. Hence, HT has a long history of being used to prevent and treat diseases and has a huge consumer market in China [1]. The demand for herbal teas has gradually increased and herbal teas are now produced on a large scale, which has led to the increased production of herbal tea residues (HTR). According to preliminary statistics, the main HT production enterprises in Guangdong can produce about 680 t HTR every day. Traditional HTR treatment methods, such as incineration and landfill, will cause many environmental problems, waste many resources, and therefore cannot meet the needs of modern intensive production [1]. These issues prompted the researchers to study the rational utilization of HTRs. HTRs are abundant in carboxylic acids, phenolic hydroxyl groups and alkoxy groups; therefore, accumulating studies have given priority to HTR as an adsorbent for heavy metal ions [2,3,4,5]. HTRs can also be used as composting materials to enhance soil fertility because of their high contents of organic matter and nitrogen [6].

Alternatively, HTR can also be used as a new type of feed resource for animal production; many nutrients and biologically active substances (Appendix A), such as flavonoids, polysaccharides, and alkaloids [7], are retained in HTRs, because not all active ingredients are decomposed during HT production. These active substances participate in a variety of physiological functions in humans and animals, including antioxidant [8], anti−inflammatory [9] and antibacterial activities [10]. Researchers found that HTR can be used as a feed additive to improve the intestinal function of fatteners [11] and strengthen the immune performance of fattening cattle under heat stress [1]. Moreover, some studies showed that tea residues, because of their high polysaccharides and alkaloids contents, could be used as feed additives to improve the meat quality of goats [12], and change the gut environment of weaned piglets [1].

The feed made from roughage via microbial fermentation is called fermented feed. The advantage of fermented feed lies in the effective degradation of anti−nutritional factors of roughage after microbial fermentation, and it can produce a great number of metabolites beneficial to animal health [13]. Therefore, in animal husbandry production, the development and promotion of fermented feed have attracted much attention.

In our paper, the HTR used in the experiment contains Chinese herbal ingredients such as *Lonicera japonica* Thunb., chrysanthemum, *Plumeria rubra*, *Mesona chinensis* Benth., *Prunella vulgaris* L. and *Glycyrrhiza uralensis* Fisch. The potential feed value of HTR has led us to develop a new feed resource for ruminant production. To date, few studies have evaluated the physiological safety of goats after adding fermented HTR (FHTR) to the diet. In this study, we hypothesized that feeding Chuanzhong black goats with FHTR instead of whole corn silage (WCS) will not adversely affect their health. This hypothesis was tested by applying a combination of serum indices and 16S rRNA high−throughput sequencing. This study was undertaken to determine the food safety of FHTR and lay the foundation for its use as a new feed resource for goats.

## 2. Materials and Methods

### 2.1. Silage Material and Preparation

FHTR was offered by Wong Lo Kat Co, Ltd. (Guangzhou, China). In total, 33 female Chuanzhong black goats (121.00 ± 4.00 days) with similar weight (9.33 ± 0.95 kg) and genetic background were used in this experiment. On the basis of the research of Xie et al., (2020) and Chinese Feeding Standards (China Standard NY/T816−2004), the experimental diets were composed of three FHTR replacement levels (0%, 15 and 30%) as substitute for WCS (labeled as CON, L15 and L30, respectively). The dietary ingredients and nutrient compositions for the trial were measured by the procedures of the Association of Official Analytical Chemists (AOAC, 2000) and are shown in Table 1. Net energy for maintenance and net energy for gain were calculated according to the method provided by the Chinese Feeding Standards (China Standard NY/T816−2004). The goats were randomly allocated to three groups, raised in captivity, and received a random diet. The experiment period lasted for 42 days. The first 7 days were used for adaptation, and the last 35 days were used for measurements. The goats were fed twice daily at 8:00 and 15:00 and provided with water ad libitum throughout the experiment.

### 2.2. Analyses of Fermentation Parameter and Chemical Composition

Blood samples were gathered from each group on days 0, 17 and 35 via jugular vein before the morning feeding, and the samples were centrifuged at 3000 r/min for 15 min. The supernatant was placed in a 2 mL sterile Eppendorf (EP) tube and stored at −80 °C for further analysis. Serum biochemical indices (albumin (ALB), total protein (TP), globulin (GLOB), total cholesterol (TC), urea (UREA), non−esterified fatty acid (NEFA), low−density lipoprotein−cholesterol (LDL−C), high−density lipoprotein−cholesterol (HDL−C) and aspartate aminotransferase (AST)) were determined using a biochemical auto−analyser (Hitachi automatic biochemical analyser 7080, Tokyo, Japan). Some oxidative stress indices were also examined. C reactive protein (CRP) was measured using radioimmunoassay reagents kits (Ketao Biotechnology Centre, Shanghai, China). Beta−hydroxybutyric acid (BHBA) was detected using the relevant commercial enzyme−linked immunosorbent assay kits (Ketao Biotechnology Center, Shanghai, China). Inhibition of hydroxyl radical (OH–), total antioxidant capacity (T−AOC), malonaldehyde (MDA) and glutathione peroxidase (GSH−Px) were detected by using the colourimetric method. Moreover, the serum samples were analyzed for tumor necrosis factor−alpha (TNF−α), interleukin−2 (IL−2), immunoglobulin A (IgA), immunoglobulin M (IgM) and immunoglobulin G (IgG) using the relevant commercial enzyme−linked immunosorbent assay kits (Ketao Biotechnology Centre, Shanghai, China).

On day 35, the fecal samples were collected from the rectum before morning feeding. Microbial DNA was extracted from the fecal samples according to the cetyltrimethylammonium bromide method, and high−throughput sequencing was performed using the PacBio Sequel platform. Nanodrop2000/2000c nucleic acid protein detector (NanoDrop Scientific Inc, Eugene, OR, USA) was used to calculate the final DNA concentration and purity, and DNA quality was verified using 1% agarose gels. The V1–V9 regions of the 16s rRNA genes were amplified using primers. Polymerase chain reaction (PCR) amplification was performed using the TransStart^®^ FastPfu DNA polymerase kit (TransGen, Biotech, Beijing, China), and all details of primers, PCR, product size and temperature are shown in Appendix A). The original sequences were initially processed by the PacBio SMRT portal. The files generated by the PacBio platform were selected. The read sequence data were compared with the reference database using the UCHIME algorithm to detect and eliminate chimeric sequences, and obtain clean reads for subsequent analysis [14,15]. The clean read sequences were analyzed using the UPARSE software and assigned at 97% sequence similarity. The operational taxonomic unit (OTU) abundance information was standardized using a sequence number standard corresponding to the sample with the least sequence. Alpha and beta diversity analyses were performed based on the homogenized data. Alpha diversity was calculated using the QIIME software, and the differences between groups were analyzed using the R software. Bacterial diversity (Ace index and observed species index) and richness (Simpson and Shannon indices) indices for assessing alpha diversity were calculated. Beta diversity is a comparative analysis that assesses the microbial community composition of different samples from all groups. Unweighted UniFrac distances were calculated using the phyloseq default script to measure Beta diversity. We examined the relationships in fecal microbiota between samples from the three experimental groups using principal component analysis (PCA). The Kyoto Encyclopedia of Genes and Genomes (KEGG) PATHWAY database is divided into seven metabolic pathway categories (metabolism; genetic information processing; environmental, information processing; cellular processes; organic systems; human diseases; drug development). Each category can be divided into level 2 (66 pathways), level 3 (metabolic pathway diagram) and level 4 (specific annotation information for each metabolic pathway diagram). Function prediction was detected by Tax4Fun, which is an R package for function prediction based on the16s Silva database [16].

### 2.3. Statistical Analysis

Excel 365 was used for the preliminary calculation and formatting of serum indices and fecal microorganism data. The MIXED statement of SAS (version 9.4; SAS Institute Inc., Cary, NC, USA) was used to calculate whether the serum indices would be affected by the interaction of diet and feeding time, the statement was set up as follows: the CLASS statement defines categorical variables, and the MODEL statement defines serum indices as dependent variables, taking diet, feeding time and diet × feeding time as independent variables. The option DDFM = KENWARDROGER was used to estimate the correction degree of freedom, the REPEATED statement defines the variance and covariance of repeated measures, the type of variance and covariance structure is an auto−regressive model (TYPE = AR(1)) and the LSMEANS statement is used to count the mean. When the interaction is not significant, a polynomial orthogonal comparison method was used to analyze the linear and quadratic relationships of diet treatment on serum indicators and fecal microorganisms.

The differences between the means of each treatment were compared by the Bonferroni method. Data were expressed as the mean and standard errors of means (SEM) in the tables. Significant differences were declared at *p* < 0.05.

## 3. Results

### 3.1. Serum Indices

Appendix A report the effect of dietary treatment on serum indices. Our finding suggested that the serum indices were not affected by the interaction between diet and feeding time. For the serum biochemical indices, UREA concentrations increased linearly with the increase of in FHTR on day 35 (*p* < 0.05). The concentrations of antioxidant capacity−related indices (MDA, T−AOC, BHBA and CRP) and immune−related indices (IgM, IgA, IgG, IL−2 and TNF−α) were not affected by the diet (*p* > 0.05). However, on day 35, OH− increased concentrations linearly and GSH−Px concentrations increased quadratically with the increase in FHTR (*p* < 0.05).

### 3.2. Sequencing Depth and Fecal Microbiota Diversity

Fecal samples were sequenced through the V1–V9 region of the 16S rRNA on the PacBio platform, and a total of 340,816 original readings were obtained. After quality control, 10,111 high−quality sequences in each sample on average were used for later analysis. Based on 97% sequence similarity, these sequences were assigned to 2709 OTUs. Group observed species (Figure 1A) indicated that the sequencing depth is sufficient to be used for the next step of data analysis. From the Venn figure, 761 of the 2709 OTUs existed corporeally in the three groups. In addition, 394, 399 and 432 unique OTUs were identified in the CON, L15 and L30 groups, respectively (Figure 1B).

As shown in Figure 1C–F, the bacterial diversity (Ace and observed species indices) and richness indices (Simpson and Shannon indices) of CON, L15 and L30 fecal microbes did not differ substantially between groups. As shown in Figure 1G, the fecal microbial communities of the CON, L15 and L30 samples were relatively concentrated, and the distances of the samples were not remarkably separated in the beta diversity PCA.

### 3.3. Composition of Fecal Microbiota at Various Taxonomic Levels

The 11 phyla above 0.1% relative abundance are listed in Table 2. The most prevalent bacteria in CON, L15 and L30 were Firmicutes (52.12%, 56.42% and 58.67%), followed by *Bacteroidetes* (37.27%, 33.17% and 31.67%) and unidentified bacteria (1.66%, 3.88% and 1.92% (Figure 2A). With respect to the genus level, the relative abundances of 14 bacteria above 0.1% are listed in Table 3. The dominant bacterial genera in CON, L15 and L30 were *Alistipes* (2.09%, 2.23% and 2.40%, respectively), *Bacteroides* (2.44%, 1.70% and 1.95%, respectively), and *Campylobacter* (1.45%, 3.80% and 1.90%, respectively) (Figure 2B). In general, diet with FHTR did not alter the order of dominant rumen bacteria. However, the relative abundance of *Phascolarctobacterium* increased quadratically (*p* = 0.048) with the increased of in FHTR. In addition, the diets had a tendency of affecting the relative abundance of *Planctomycetes* (*p* = 0.060).

### 3.4. Function Prediction of Fecal Microbiota

In the current study, Tax4fun was used for predicting the fecal microbial function of the different treatment groups (Figure 3). We mainly focused on the effect of FHTR on the level 2 KEGG function analysis of fecal microorganisms. As shown in Appendix A and Figure 4, the top five predicted functions were Carbohydrate_metabolism, Replication_and_repair, Membrane_transport, Amino_acid_metabolism and Translation. Diet with FHTR did not change the order of these pathways. The PCA of the predicted pathways was performed and the result is shown in Figure 5. The samples were relatively concentrated, which indicated that adding FHTR to the diet did not alter the fecal microbial function.

## 4. Discussion

### 4.1. Serum Indices

Homeostasis in the animal body and the damaged state of internal organs can be presented by serum indices [17]. Cell metabolism will produce a sequence of reactive oxygen species (ROS), which can be converted into hydrogen peroxide (H_2_O_2_) and hydroxyl radicals (OH–) [18]. The oxidation of biomolecules in organisms is mediated by OH–formation. OH– is the most effective oxidant that can impair the function of proteins by affecting the conduction of receptors and antibody symbols [19,20]. Additionally, it can induce the destruction of deoxyribose and ribose, causing the oxidative damage of DNA [21,22]. Oxidative stress will occur in the body when ROS production surpasses the limited capacity of the cellular antioxidant system. Fortunately, organisms will restore homeostasis through the antioxidant defense system, including enzyme scavenger and non−enzyme molecules [23]. GSH−Px is an important peroxide that decomposes enzymes and is widely present in the body. Its active center is selenocysteine, which can diminish toxic peroxides to non-toxic hydroxyl compounds and block further damage to organisms [24]. The antioxidant active substances in the diet can influence the oxidation status of the plasma, muscle and liver [25]. On day 35, the L30 group had higher OH– and GSH−Px concentrations than the other groups. In a previous study, 36 antioxidant active substances of HT were reported using high−performance liquid chromatography (HPLC)–tandem mass spectroscopy and HPLC–evaporative light scattering detection [26]. The results imply that FHTR contains a variety of antioxidant active substances (flavonoids and polyphenols) that led to these changes. Therefore, we tentatively concluded that FHTR is harmless to Chuanzhong black goats and has the potential to be used as an animal feed.

### 4.2. Fecal Microbiota

A mounting number of studies have shown that the composition of gut microorganism is closely related to the health of the host [27]. In our study, we analyzed the changes in the diversity of fecal microbiota by adding FHTR to the diet through high−throughput platform sequencing. We found that adding FHTR to the diet had no adverse effect on the alpha structures of fecal microorganisms and did not change the dominant phylum of the fecal microbiota. At the phylum level, the dominant bacteria were *Bacteroides* and *Firmicutes*, which is consistent with previous research [28,29]. The relative abundance of *Phascolarctobacterium* notably increased in the L15 group compared with the other groups. *Phascolarctobacterium* inhibits the growth of Clostridium difficile by consuming succinic acid and thus reduces *C. difficile* infection [30]. In addition, some scholars explored the effect of loquat leaf polysaccharides on gut microorganisms and found that loquat leaf polysaccharides can remarkably increase the relative abundance of *Phascolarctobacterium* [31]. *Phascolarctobacterium* may play a central part in host immunity with an unfamiliar mechanism.

The diets affected the relative abundance of Planctomycetes. Anaerobic ammonium oxidation (Anammox) plays an important role in the global carbon and nitrogen cycle using carbon dioxide or carbonate as the carbon source, ammonium salt as electron donor and nitrite/nitrate as the electron acceptor [32]. The bacteria with anammox function are largely derived from *Planctomycetes* [33,34]. To date, Planctomycetes has 29 genera, most of which represent merely one species [35]. Ruminants have a unique regulation mechanism of endogenous UREA recycling, which permits UREA nitrogen to be recycled in a constant way [36,37,38]. To date, much of the research on *Planctomycetes* has focused on wastewater treatment and nitrogen removal from farmland soil systems, and few have focused on ruminants [39,40,41,42,43]. Perhaps this bacterial phylum affects the recovery of endogenous UREA in ruminants through a complicated regulatory mechanism.

Tax4fun classifies OTUs according to the SILVA database. Based on the classification results, the 16S copy number is standardized according to NCBI genome annotation. Eventually, the linear relationship between SILVA classification and KEGG database prokaryotic classification is constructed to realize the function prediction of microorganisms [44]. In the current study, our findings suggested that diet with FHTR did not change fecal microbial function.

The competitiveness of this study lies in the fact that we measured a large number of serum indices and combined the indices with 16S rRNA gene sequencing, which enabled us to understand the safety of FHTR and determine whether it could be used as animal feed. We did not follow the animals for a long time, and few serum indices were different in this study, owing to certain objective conditions. Therefore, we cannot infer whether supplementing FHTR with the diet improves the antioxidant function of Chuanzhong black goats. In spite of some limitations, our findings suggest that adding FHTR to the diet did not affect the health of Chuanzhong black goats and could open up a new method of FHTR application.

## 5. Conclusions

In this work, serum indices and high−throughput sequencing were combined to evaluate whether FHTR has potential as an animal feed. The results showed that adding FHTR to the diet can increase the concentrations of serum GSH−Px and OH– in Chuanzhong black goats and has no adverse effects on other serum indices. In addition, FHTR did not change the alpha structures and major functions of fecal microorganisms. In conclusion, FHTR has the potential to be used as feed for Chuanzhong black goats.

## Figures and Tables

**Figure 1 microorganisms-10-01228-f001:**
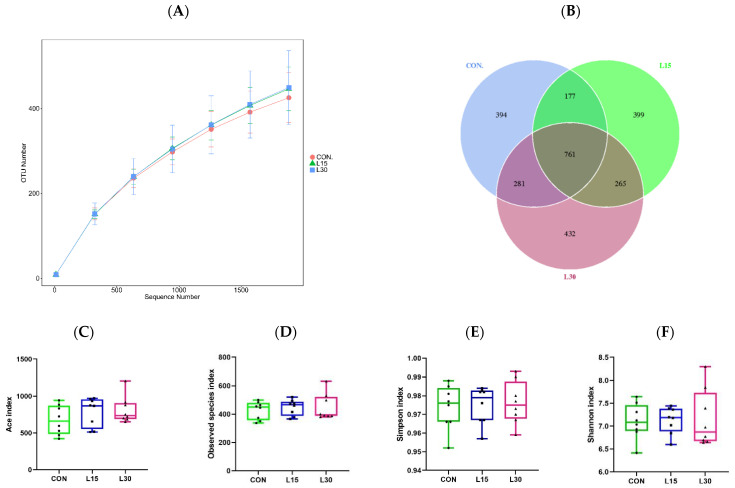
Sequencing depth and fecal microbiota diversity. (**A**) Group observed species; (**B**) Venn figure; (**C**) Ace index; (**D**) Observed species index; (**E**) Simpson index; (**F**) Shannon index; and (**G**) OUT, operational taxonomic unit; PCA, principal component analysis; CON, 0% FHTR in the diet; L15, 15% FHTR in the diet; L30, 30% FHTR in the diet.

**Figure 2 microorganisms-10-01228-f002:**
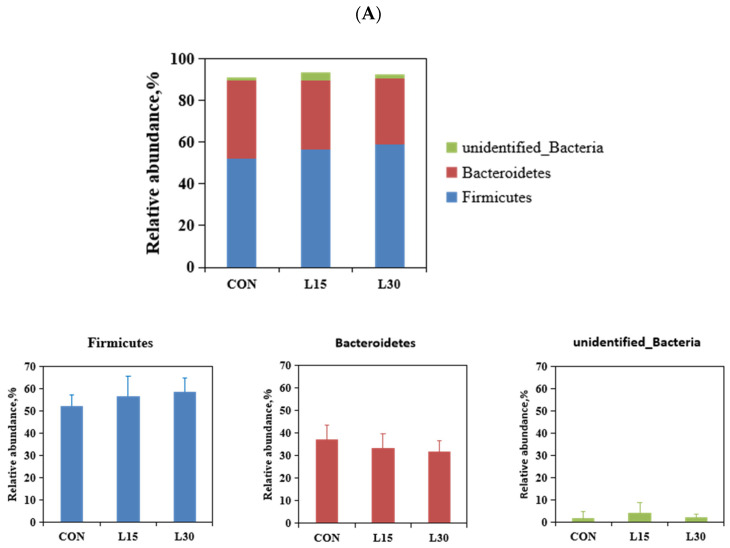
Effect of fermented herbal tea residue silage (FHTR) on fecal microbiota composition. The fecal microbiota composition of goats in CON, L15 and L30 group at (**A**) phylum and (**B**) genus level.

**Figure 3 microorganisms-10-01228-f003:**
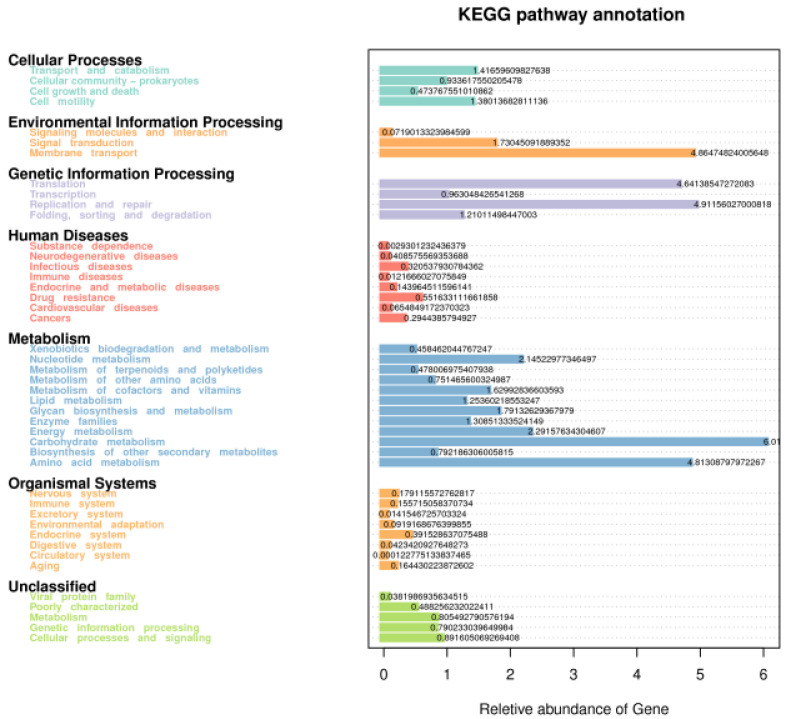
Statistical graph of the gene prediction function results of fecal microorganism via Tax4fun. CON, 0% FHTR in the diet; L15, 15% FHTR in the diet; L30, 30% (FHTR) in the diet.

**Figure 4 microorganisms-10-01228-f004:**
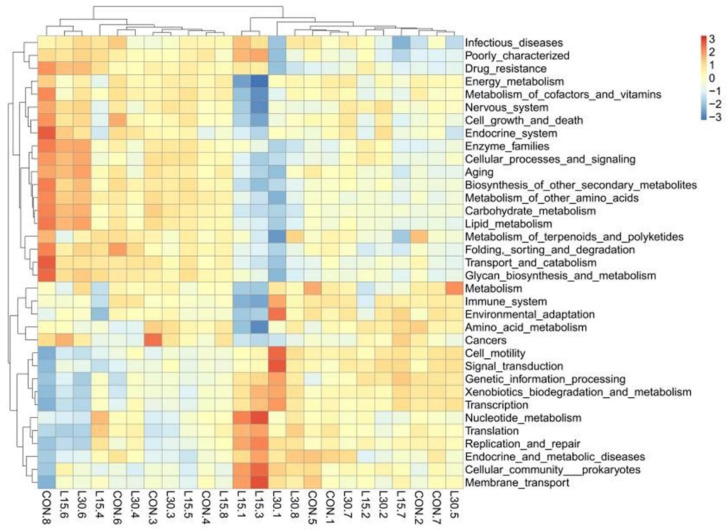
Cluster heatmap of the relative abundance of fecal microorganisms in the functional analysis via Tax4fun. CON, 0% FHTR silage in the diet; L15, 15% FHTR in the diet; L30, 30% FHTR in the diet.

**Figure 5 microorganisms-10-01228-f005:**
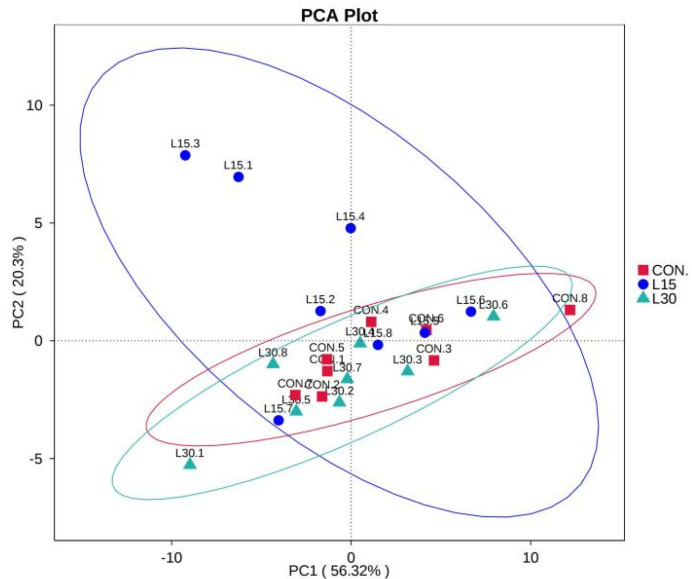
PCA of level 2 functional analysis of fecal microorganisms via Tax4fun. CON, 0% FHTRin the diet; L15, 15% FHTR in the diet; L30, 30% FHTR in the diet.

**Table 1 microorganisms-10-01228-t001:** Composition and nutrient levels of experimental diets for Chuanzhong black goats (dry matter basis).

Items	Dietary Treatment		
	CON	L15	L30
Ingredient, %			
Whole plant corn silage	76.2	64.8	53.3
Fermented herbal tea residue	0	11.4	22.9
Peanut seedling feed	19	19	19
Soya bean meal	1.2	1.2	1.2
Alfalfa	0.47	0.47	0.47
Mountain flour	0.1	0.1	0.1
Salt	0.05	0.05	0.05
Nutritional ingredient, %			
DM	30.84	33.45	31.33
CP	11.68	12.13	11.79
EE	0.01	0.01	0.02
Ash	0.13	0.12	0.12
NDF	69.23	69.93	68.76
ADF	52.21	53.83	51.49
Ca	0.9	0.91	1.04
P	0.44	0.44	0.41

CON, 0% fermented herbal tea residue (FHTR) in the diet, L15, 15% FHTR in the diet; L30, 30% FHTR in the diet. DM, dry matter; CP, crude protein; EE, ether extract; NDF, neutral detergent fiber; ADF, acid detergent fiber; Ca, calcium; P, phosphorus.

**Table 2 microorganisms-10-01228-t002:** Effects of different proportions of FHTR on the bacteria composition of Chuanzhong black goats at the phylum level (relative abundance ≥ 0.1%).

Items	Diet			SEM	*p*-Value	
	CON	L15	L30		Line	Quad
Firmicutes	52.12	56.42	58.67	1.589	0.100	0.760
Bacteroidetes	37.27	33.17	31.67	1.326	0.090	0.639
unidentified_Bacteria	1.66	3.88	1.92	0.746	0.889	0.202
Spirochaetes	2.44	1.70	2.77	0.703	0.852	0.564
Tenericutes	2.72	2.02	2.36	0.274	0.600	0.390
Proteobacteria	0.88	0.46	0.57	0.155	0.438	0.436
Melainabacteria	1.18	1.17	1.01	0.142	0.637	0.801
Deferribacteres	0.44	0.15	0.19	0.098	0.311	0.439
Elusimicrobia	0.03	0.13	0.05	0.038	0.835	0.252
Lentisphaerae	0.40	0.19	0.22	0.052	0.163	0.292
Planctomycetes	0.28	0.23	0.14	0.030	0.060	0.788

SEM, standard error of mean. Means with different letters in the same row (a–c) differ significantly (*p* < 0.05) and means with no letters or the same letters are not significantly different. CON, 0% FHTR in the diet; L15, 15% FHTR in the diet; L30, 30% FHTR in the diet. *p*−values are for linear and quadratic orthogonal contrasts for diet.

**Table 3 microorganisms-10-01228-t003:** Effects of different proportion of fermented herbal tea residue on bacterium composition at general level of Chuanzhong black goats (relative abundance ≥ 0.1%).

Items	Diet			SEM	*p*-Value	
	CON	L15	L30		Line	Quad
*Alistipes*	2.09	2.23	2.40	0.225	0.592	0.974
*Bacteroides*	2.44	1.70	1.95	0.220	0.373	0.298
*Campylobacter*	1.45	3.80	1.90	0.751	0.812	0.199
*Roseburia*	1.00	0.64	1.85	0.277	0.210	0.183
*Anaerovibrio*	0.10	0.56	0.98	0.237	0.140	0.963
*unidentified_Ruminococcaceae*	1.55	1.85	2.30	0.179	0.095	0.853
*Phascolarctobacterium*	1.70 ^a^	2.24 ^b^	1.57 ^a^	0.145	0.697	0.048
*Mucispirillum*	0.44	0.15	0.19	0.098	0.311	0.439
*Tyzzerella*	0.25	0.37	0.47	0.072	0.216	0.932
*unidentified_Clostridiales*	0.57	0.35	0.51	0.065	0.709	0.167
*Anaerosporobacter*	0.44	0.4	0.31	0.073	0.480	0.870
*Candidatus_Soleaferrea*	0.61	0.56	0.62	0.040	0.948	0.523
*Anaerovorax*	0.27	0.23	0.21	0.027	0.381	0.955
*Turicibacter*	0.19	0.27	0.18	0.035	0.881	0.307

SEM, standard error of mean. Means with different letters in the same row (a,b) differ significantly (*p* < 0.05) and means with no letters or the same letters are not significantly different. CON, 0% FHTR in the diet, L15, 15% FHTR in the diet; L30, 30% FHTR in the diet. *p*−values are for linear and quadratic orthogonal contrasts for diet.

## Data Availability

The raw sequencing data from this study is deposited in the Genome Sequence Archive in Beijing Institute of Genomics (BIG) Data Centre (https://bigd.big.ac.cn/, accessed on 28 December 2021), under accession number: CRA005724.

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
