# Peer review of "Effects of Fermented Herbal Tea Residue on Serum Indices and Fecal Microorganisms of Chuanzhong Black Goats"

_microorganisms, 2022, doi:10.3390/microorganisms10061228_

Round 1
Reviewer 1 Report
Introduction
Please provide a table (or supplementary material) with all beneficial substances found in herbal tea according to the international literature.
Materials and methods
2.1. Please indicate how you knew that animals were of the same genetic background.
2.2. Was the biochemical analyser calibrated for use in goats?
All the details of the PCR, primers, product size, temperature etc. should be included in a table, not in the text.
Results
Table 2. Too detailed, please move to supplementary material. Also, please include the reference values for these parameters in goats.
Tables 3 and 4. Too detailed, please move to supplementary material.
3.2. “We calculated bacterial diversity (Ace and observed species index) (Figure 1-C, D) 191
and richness (Simpson and Shannon index) indices …” These are methods, please move into relevant section of the manuscript.
Same also in other passages in results.
Discussion
Please divide in two sub-sections
Also, some significant references are missing, please add.
Reviewer 2 Report
There is need to take note of the following in abstract
line 15. was lasted should be lasted
line 17. increase should be increased. i.e. use appropriate past tense. Check this throughout the manuscript.
line 19. FHTR no significantly ... should be FHTR had no significant effect....
Round 2
Reviewer 1 Report
The following two points from the initial review were not answered.
Please provide clear and detailed answers to these points.
Point 2: 2.1. Please indicate how you knew that animals were of the same genetic background.
Point 3: 2.2. Was the biochemical analyser calibrated for use in goats?
Author Response
Response to Reviewer 1 Comments
Point 2: 2.1. Please indicate how you knew that animals were of the same genetic background.
Response 2: In this study, three 33 female Chuanzhong black goats were selected from the same breeding farm, with the same weight (9.33 ± 0.95 kg), age (121.00 ± 4.00 days) and genetic background.The breeding farm data showed that the fathers of the 33 female Chuanzhong black goats used in this experiment were the same ram.
Point 3: 2.2. Was the biochemical analyser calibrated for use in goats?
Response 3: Yes, our laboratory often does goat-related tests, and the reference values of the biochemical analyzer used in the test are calibrated before being used for the determination of goat serum biochemical indicators.
Biochemical analyzers have been used extensively in goats. According to reports, the serum biochemical indexes of white cashmere goats can be measured by biochemical analyzer to explore the effect of feeding mulberry leaves on goats(Wang et al.,2020). In addition, Biochemical analyzers have long been used in goat experiments to identify goats' health status.(Clark et al.,1999).
References:
- Wang, Y.Y.; Shen, Q.M.; Zhong, S.; Chen, Y.L.; Yang, Y.X. Comparison of Rumen Microbiota and Serum Biochemical Indices in White Cashmere Goats Fed Ensiled or Sun-Dried Mulberry Leaves. Microorganisms. 2020, 8, 10.3390/microorganisms8070981
- Clark, P.; Swenson, C.L.; Osborne, C.A.; Ulrich, L.K. Calcium oxalate crystalluria in a goat. JAVMA-J AM VET MED A. 1999, 215, 77-8.